Ship noise extends to frequencies used for echolocation by endangered killer whales

Veirs Scott 1 scott@beamreach.org
Veirs Val 2
Wood Jason D. 3
1 Beam Reach Marine Science and Sustainability School , Seattle, WA , United States
2 Department of Physics, Colorado College , Colorado Springs, CO , United States
3 SMRU Consulting , Friday Harbor, WA , United States
Johnson Magnus
Electronic publication date: 2016 Feb 2
Publication date: 2016
Volume: 4
Electronic Location ID: e1657
Received 2015 Sep 29; Accepted 2016 Jan 13
Copyright: ©2016 Veirs et al.
Copyright year: 2016
Copyright holder: Veirs et al.
License: This is an open access article distributed under the terms of the Creative Commons Attribution License, which permits unrestricted use, distribution, reproduction and adaptation in any medium and for any purpose provided that it is properly attributed. For attribution, the original author(s), title, publication source (PeerJ) and either DOI or URL of the article must be cited.
License URL: https://creativecommons.org/licenses/by/4.0/

Keywords: Noise, Hydrophone, Killer whale, Orca, Odontocete, Marine mammal, Ship, Pollution, Acoustics, Bioacoustics

Funding: Northwest Fisheries Science Center NOAA’s Western Regional Center Washington State Parks Chuck Greene of Cornell University Funding for the Salish Sea hydrophone network came via Brad Hanson of the Northwest Fisheries Science Center and Lynne Barre of NOAA’s Western Regional Center, Washington State Parks, and Chuck Greene of Cornell University. The funders had no role in study design, data collection and analysis, decision to publish, or preparation of the manuscript.

==============================
Combining calibrated hydrophone measurements with vessel location data from the Automatic Identification System, we estimate underwater sound pressure levels for 1,582 unique ships that transited the core critical habitat of the endangered Southern Resident killer whales during 28 months between March, 2011, and October, 2013. Median received spectrum levels of noise from 2,809 isolated transits are elevated relative to median background levels not only at low frequencies (20–30 dB re 1 µPa2/Hz from 100 to 1,000 Hz), but also at high frequencies (5–13 dB from 10,000 to 96,000 Hz). Thus, noise received from ships at ranges less than 3 km extends to frequencies used by odontocetes. Broadband received levels (11.5–40,000 Hz) near the shoreline in Haro Strait (WA, USA) for the entire ship population were 110 ± 7 dB re 1 µPa on average. Assuming near-spherical spreading based on a transmission loss experiment we compute mean broadband source levels for the ship population of 173 ± 7 dB re 1 µPa 1 m without accounting for frequency-dependent absorption. Mean ship speed was 7.3 ± 2.0 m/s (14.1 ± 3.9 knots). Most ship classes show a linear relationship between source level and speed with a slope near +2 dB per m/s (+1 dB/knot). Spectrum, 1/12-octave, and 1/3-octave source levels for the whole population have median values that are comparable to previous measurements and models at most frequencies, but for select studies may be relatively low below 200 Hz and high above 20,000 Hz. Median source spectrum levels peak near 50 Hz for all 12 ship classes, have a maximum of 159 dB re 1 µPa2/Hz @ 1 m for container ships, and vary between classes. Below 200 Hz, the class-specific median spectrum levels bifurcate with large commercial ships grouping as higher power noise sources. Within all ship classes spectrum levels vary more at low frequencies than at high frequencies, and the degree of variability is almost halved for classes that have smaller speed standard deviations. This is the first study to present source spectra for populations of different ship classes operating in coastal habitats, including at higher frequencies used by killer whales for both communication and echolocation.

Introduction

Commercial ships radiate noise underwater with peak spectral power at 20–200 Hz (Ross, 1976). Ship noise is generated primarily from propeller cavitation, propeller singing, and propulsion or other reciprocating machinery (Richardson et al., 1995; Wales & Heitmeyer, 2002; Hildebrand, 2009). The dominant noise source is usually propeller cavitation which has peak power near 50–150 Hz (at blade rates and their harmonics), but also radiates broadband power at higher frequencies, at least up to 100,000 Hz (Ross, 1976; Gray & Greeley, 1980; Arveson & Vendittis, 2000). While propeller singing is caused by blades resonating at vortex shedding frequencies and emits strong tones between 100 and 1,000 Hz, propulsion noise is caused by shafts, gears, engines, and other machinery and has peak power below 50 Hz (Richardson et al., 1995). Overall, larger vessels generate more noise at low frequencies (<1,000 Hz) because of their relatively high power, deep draft, and slower-turning (<250 rpm) engines and propellers (Richardson et al., 1995).

This low-frequency energy from ships is the principal source of ambient noise within the deep ocean from approximately 5–1,000 Hz (Wenz, 1962; Urick, 1983; National Research Council et al., 2003). Growth of the global shipping fleet and possibly the average size of ships has raised deep-ocean ambient noise levels in low-frequency bands near 40 Hz by up to 20 dB relative to pre-industrial conditions (Hildebrand, 2009) and 8–10 dB since the 1960s (Andrew et al., 2002; McDonald, Hildebrand & Wiggins, 2006).

As these ships enter shallow waters and traverse the estuarine habitat typically occupied by major ports, the noise they radiate may impact coastal marine life. Since many marine mammals rely on sound to find prey, moderate social interactions, and facilitate mating (Tyack, 2008), noise from anthropogenic sound sources like ships can interfere with these functions, but only if the noise spectrum overlaps with the hearing sensitivity of the marine mammal (Southall et al., 2007; Clark et al., 2009; Hatch et al., 2012).

Hearing sensitivity isn’t yet characterized in Mysticetes (baleen whales), but based on their signals they are likely most sensitive at frequencies 10–10,000 Hz and therefore constitute a low-frequency functional hearing group (Southall et al., 2007). They typically emit signals with fundamental frequencies well below 1,000 Hz (Cerchio, Jacobsen & Norris, 2001; Au et al., 2006; Munger et al., 2008) although non-song humpback signals have peak power near 800 and 1,700 Hz (Stimpert, 2010) and humpback song harmonics extend up to 24,000 Hz (Au et al., 2006).

The frequency overlap of peak power in ship noise and baleen whale signals (and inferred maximum hearing sensitivity) is verified by observed behavioral and physiological responses of mysticetes to ship noise. As examples, the probability of detecting a blue whale D call increases in ship noise, suggesting a Lombard effect (Melcon et al., 2012) and Rolland et al. (2012) found decreased stress levels in North Atlantic right whales when ship noise was absent.

The potential impacts of ship noise can be assessed more confidently in Odontocetes (toothed whales) because they constitute mid-frequency or high-frequency functional hearing groups (Southall et al., 2007) in which auditory response curves have been obtained for many species. These curves show maximum auditory sensitivity near the frequencies where toothed whale signals have peak power (Mooney, Yamato & Branstetter, 2012; Tougaard, Wright & Madsen, 2014)—at about 1,000–20,000 Hz for social sounds and 10,000–100,000 Hz or higher for echolocation.

Southern Resident killer whales (SRKWs) represent an endangered toothed whale species that inhabits an urban estuary in which shipping traffic is common and is very well characterized bioacoustically. Their auditory sensitivity, extrapolated from captive killer whales (Hall & Johnson, 1972; Szymanski et al., 1999), peaks at 15,000–20,000 Hz—a frequency range that overlaps with the upper range of their vocalizations and the lower range of their echolocation clicks. SRKW calls have fundamental frequencies at 100–6,000 Hz with harmonics extending up to 30,000 Hz (Ford, 1987). Their echolocation clicks are likely similar to those of salmon-eating northern resident killer whales which have a 40,000 Hz bandwidth and a mean center frequency of 50,000 Hz (Au et al., 2004). SRKWs whistle between 2,000 and 16,000 Hz (Riesch, Ford & Thomsen, 2006) with a mean dominant frequency of 8,300 Hz (Thomsen, Franck & Ford, 2000).

Behavioral responses to boat (as opposed to ship) noise have been documented in toothed whales, including SRKWs. For example, bottlenose dolphins whistle (at 4,000–20,000 Hz) less when exposed to boat noise at 500–12,000 Hz (Buckstaff, 2004) and Indo-Pacific bottlenose dolphins lower their 5,000–10,000 Hz whistle frequencies when noise is increased by boats in a band from 5,000 to 18,000 Hz (Morisaka et al., 2005). For every 1 dB increase in broadband underwater noise (1,000–40,000 Hz) associated with nearby boats, SRKWs compensate by increasing the amplitude of their most common call by 1 dB (Holt et al., 2009).

While the frequencies used by toothed whales are well above the peak power frequencies of ships, multiple lines of evidence suggest that ship noise spectra extend or should be expected to extend to higher frequencies. Laboratory experiments with cavitation and previous studies of submarines, torpedoes, and ships indicate that ship noise may extend as high as 160,000 Hz at the source.

Experiments confirm that cavitation generates high frequency noise up to at least 100,000 Hz (Wenz, 1962). Cavitation noise from spinning rods and water jets has spectral power that rises through low frequencies at a rate of 40 dB/decade to a peak near 1,000 Hz and thereafter descends at −20 dB/decade (Mellen, 1954; Jorgensen, 1961). Noise from foil cavitation also has peak spectral power at 1,000 Hz, as well as a secondary peak at 31,000 Hz (Blake, Wolpert & Geib, 1977). In the vicinity of the higher peak, 1/3-octave levels increase about 10 dB upon cavitation inception (Blake, Wolpert & Geib, 1977).

World War II studies of ship noise, particularly measurements of thousands of transits of hundreds of ships of all types, identified propeller cavitation as the dominant source of noise radiated by ships, including at high frequencies (Dow, Emling & Knudsen, 1945). In reviewing these studies Ross (1976) and Urick (1983) noted that increases of >40 dB in the 10,000–30,000 Hz band were diagnostic of cavitation inception on accelerating twin-screw submarines and Urick (1983) attributed a 1 dB/knot (2 dB per m/s) rise in torpedo spectrum levels from 10,000 to 75,000 Hz to propeller cavitation.

More recently, cavitation has been implicated in ship noise measurements made at close range (<1,000 m) which show levels between 1,000 and 60,000 Hz that not only are significantly above background levels, but also rise with increased ship speed faster than at lower frequencies (Arveson & Vendittis, 2000; Kipple, 2002; Hermannsen et al., 2014). Even when portions of the high-frequency energy are excluded, broadband source levels of cavitating propellers are high. Erbe & Farmer (2000) reported median broadband (100–20,000 Hz) source levels for an icebreaker with a cavitating propeller of 197 dB re 1 µPa @ 1 m.

In the open ocean or on the outer continental shelf far from shipping lanes high-frequency noise radiated by a ship will be absorbed within about 10 km (Erbe & Farmer, 2000), often before reaching a species of concern. In urban estuaries, however, marine mammals are exposed to noise from ships at ranges of 1–10 km routinely, and less than 100 m occasionally. For example, SRKWs frequently transit Haro Strait within 10–300 m of the shoreline at Lime Kiln Point where they are about 2 km from the center of the northbound (nearest) shipping lane (Fig. 1). Since the absorption rate is only about 3 dB/km at 20,000 Hz, compared to 30 dB/km at 100,000 Hz (Francois & Garrison, 1982), ship noise near 20,000 Hz (where SRKWs are most sensitive) in such close quarters may retain the potential to mask echolocation clicks, as well as other high-frequency signals.

Figure 1 Study site map.

Inset regional map shows the study area (black rectangle) and shipping lanes (in red) leading to the major ports of the Salish Sea. The 240° bearing (gray arrow) extends from the Lime Kiln hydrophone (gray circle) through the northbound shipping lane. Bathymetric contours (50 m) show that Haro Strait is a steep-sided 200–300 m-deep channel. Sound projection locations (black dots) are sites used for the transmission loss experiment.

Figure 2 Comparison of source levels from different studies for various classes of ships.

Broadband source level (SL) statistics for each ship class juxtaposed with results from recent studies of comparable classes. Bold horizontal lines are medians; gray box hinges are 25% and 75% quantiles; gray whiskers extend to the value that is most distant from the hinge but within 1.5 times the inter-quartile range (distance between the 25% and 75% quantiles); red dots are mean values from Table 2. Each encircled letter B represents a mean from Bassett et al. (2012); blue vertical bars represent means from McKenna et al. (2012) with the container ship estimate of McKenna, Wiggins & Hildebrand (2013) labeled McKenna; black vertical bars represent estimates from Kipple (2002) and Arveson & Vendittis (2000).

In an environment where SRKWs may already be food-stressed (Ayres et al., 2012) due to reduced populations of their primary prey—Chinook salmon (Hanson et al., 2010)—echolocation masking could have grave population-level consequences. The potential impacts of ship noise on foraging efficiency may be compounded by simultaneous masking of communication calls, some of which may help coordinate foraging or prey sharing (Ford & Ellis, 2006). One case study has suggested that ship noise may reduce foraging efficiency by 50% in Curvier Beaked whales (Aguilar Soto et al., 2006).

Motivated by the possible impacts of ship noise on odontocetes and the scarcity of ship noise measurements made at close range over the full range of frequencies used by SRKWs, we endeavor to estimate source spectrum levels up to 40,000 Hz for a wide variety of ships from measurements made at a range of less than a few kilometers. Our primary objective is to characterize ship noise at higher frequencies, specifically those important to killer whales. A secondary objective is to compare our results with previous studies in order to understand consistencies and possible biases in field measurements of ship noise.

Methods

Our study site is an area of the inland waters of Washington State and British Columbia known as the Salish Sea. This urban estuary hosts the commercial shipping ports of Vancouver, Seattle, and Tacoma (see Fig. 1).

Shipping traffic primarily associated with Vancouver—about 20 large (>65 feet or 19.8 m) vessels per day (Veirs & Veirs, 2006)— transits Haro Strait, the core of the summertime habitat of the SRKWs (Hauser et al., 2007). Each ship typically raises sound pressure levels1near the shoreline about 20 dB re 1 µPa (RMS, 100–15,000 Hz) above background levels to about 115 dB for approximately 20 min/transit (Veirs & Veirs, 2006). We define ships as all vessels with overall length (LOA) greater than 65 feet (19.8 m); the remaining, shorter vessels (boats) are not characterized in this study.

We measured underwater noise radiated by these ships, collecting data continuously during 28 months between March 7, 2011, and October 10, 2013, except for occasional 1–2 day interruptions caused by power outages. About 3.5 months of data were excised due to systematic noise caused during equipment repairs made between July 22, 2011, and November 9, 2011. Consequently, we sampled every month of the year at least twice.

Study site

We deployed a calibrated hydrophone 50 m offshore of the lighthouse at Lime Kiln State Park in which The Whale Museum and Beam Reach maintain an acoustic observatory as part of the Salish Sea Hydrophone Network (orcasound.net). Midway along the west side of San Juan Island, Lime Kiln lighthouse sits on a point near the center of the summertime habitat of the SRKWs (Fig. 1). While the killer whales sometimes swim directly over the hydrophone location, they more typically transit the site 100–300 m offshore where received levels of noise from the shipping lanes would be somewhat higher than those recorded in this study.

The hydrophone was secured to a PVC pipe projecting vertically from a cement-filled tire resulting in a position 1 m above the bottom at a depth of 8 m (below mean lower low water). A cable protected by irrigation pipe secured in the inter- and sub-tidal zones brought the signal to recording hardware within the lighthouse and also housed a saltwater ground wire that helped reduce system noise.

The local bathymetry on a transect perpendicular to the shoreline (240° bearing) and running from the hydrophone to the northbound shipping lane descends to deep (>200 m) water within 300 m of the shoreline. The nearshore region (<150 m from shore) has a substrate of boulders and gravel covered with marine vegetation and descends at a slope of about 20°. Further from shore the bottom descends at a slope of about 45°.

Relative to the northbound shipping lane the hydrophone position is 1.3 km from the eastern edge, 2.25 km from the center of the lane, and about 3.7 km from the center of the traffic separation zone. A histogram of the range to all ships in our database shows peaks at 2.3 and 5.0 km, corresponding with the middle of the north- and south-bound lanes, respectively.

Data acquisition

We made audio recordings of the signal from a Reson TC4032 hydrophone installed with a differential output (sensitivity of −164 ± 3 dB re 1 V/µPa from 5 to 125,000 Hz) that was amplified and then digitized by a MOTU Traveller sampling at 192,000 Hz with 16 bits per sample. The maximum signal that could be recorded without clipping was 140 dB.

A Windows XP computer analyzed and archived the recorded signal. We calibrated the recording system with the analog output of an Interoceans 902 (acoustic listening calibration system) while a ship was passing the lighthouse, thereby converting the samples to decibels (dB) referenced to 1 µPa (hereafter dB re 1 µPa). This procedure was carried out occasionally to check and make minor changes in the Reson calibration constant during the 28 month study period.

A Python program analyzed the digitized hydrophone signal. The program continuously computed running 2-second mean square voltage levels. Each hour the program archived the 2-second recordings that yielded the minimum and maximum averages. We used the minimum files to determine background noise levels.

Generally, all commercial ships over 300 tons are required to use the Automatic Identification System (AIS) to broadcast navigational data via VHF radio. The AIS carriage requirements of the US Coast Guard (33 CFR 164.46) and Canada within a vessel traffic service area like Haro Strait mean that some fishing and passenger vessels may be underrepresented in our data set. Each AIS-equipped ship transmits at least its identification number, location, course, and speed a few times each minute. The typical range over which these transmissions are detected is 45 km.

The Python program scanned the binary output of an AIS receiver (Comar Systems AIS-2-USB) located in the lighthouse. For each transmission received, the location of the ship was used to calculate its range (R) from the hydrophone. When R was less than 4 nautical miles (7.4 km), the program recorded the broadband received level every 0.5 nautical mile (926 m) as the ship approached and departed. When the ship crossed a line perpendicular to shore (at an azimuth angle of 240° true, see Fig. 1), the Python program stored a 30-second WAV file, the date and time, and the decoded ship metadata (ship ID number, range, speed over the ground (SOG), and course over the ground). Given the orientation of the northbound shipping lane, this procedure made it likely that we recorded the starboard beam aspect noise levels of each isolated ship near the closest point of approach. Finally, the program calculated the calibrated broadband received level using the Reson calibration constant and the RMS amplitude of the 30-second file.

To maximize the detection of any high-frequency signal generated by passing ships, and to reduce the spatial extent of our transmission loss experiment, we elected to compute source levels for only the closer, northbound portion of the traffic in Haro Strait. Southbound traffic was recorded, counted, and archived, but is not included in this analysis. For the northbound traffic presented herein, the mean and standard deviation of R is 2.30 ± 0.39 km, and the minimum and maximum R are 0.95 km and 3.65 km, respectively.

Data analysis

Isolation and identification

Archived WAV files and associated metadata were analyzed with a C++ program developed in the platform-independent Qt environment (qt-project.org). To measure the noise radiated by an individual ship, rather than multiple ships, the program used the AIS data to detect acoustically-isolated ships. A ship was deemed isolated if the previous and subsequent ships were at least 6 nautical miles (11.1 km) away from the hydrophone when the WAV file was recorded. It is only at closer range that human listeners can detect ship noise above ambient levels.

For each isolated ship, the program used the ship’s identification (Maritime Mobile Service Identity, or MMSI) number to look up details about the ship from online web sites such as the Marine Traffic network (www.marinetraffic.com). These metadata, saved in a MySQL database, include (when available): MMSI, ship name, ship type, year built, length, breadth, dead weight, maximum and mean speed, flag, call sign, IMO, draft, maximum draft, and photographs.

We simplified 41 ship type categories returned from online queries into 12 general ship classes: bulk carrier (includes ore carriers); container; tug (includes multi-purpose offshore, pusher tug, and tender); cargo (includes other cargo, heavy lift, wood chip carrier); vehicle carrier (includes all roll-on roll-offs); tanker (includes crude oil, oil product, oil/chemical, chemical, and product tankers); military (includes Coast Guard, search and rescue); fishing (includes fish carrier, factory, fishing, fishing vessel, and trawler); passenger (includes cruise ships and ferries); miscellaneous (includes cable layer, reserved, unspecified, and well-stimulation); pleasure craft (includes sailing vessels, motor yacht, and yachts); and research.

Received levels

From each isolated ship’s WAV file the RMS power spectral density (PSD) was calculated using a Fast Fourier Transform averaged over the 30-second duration of the file (Nyquist frequency of 96,000 Hz; 16,384 (214) sample overlapping Bartlett window). The bandwidth of each of the 8,192 frequency bins was 11.5 Hz. These RMS PSD (per Hz) values were calibrated by requiring that the integral of the PSD equal the calibrated broadband level associated with each WAV file. The resulting power spectral densities we call the total received spectrum levels.

The total received spectrum level is a composite of the power that originated from the ship and the power associated with the background noise at the time of the ship passage. To enable estimation of the background level at the time of ship passage we continuously observed 2-second sound samples, saving the lowest power 2-second sample every hour.

The subtraction of the estimated background received level (RLB) from the total received spectrum level (RLT) to determine the received spectrum level associated with the ship (RLS) is based on the fact that when two or more waves pass at once, the pressure on the hydrophone (P) is the sum of the instantaneous pressure from each wave. The power that we calculate is proportional to the square of the pressure on the hydrophone and is represented in decibels. These relationships apply both for the power at individual frequencies (PSD) and the total power (PwrT) integrated over all frequencies.

Following the nomenclature of Erbe (2011), (1) PwrTt=kPSt+PBt2,

where k is a constant dependent on the construction of the hydrophone and t is time. Averaging over the 30 s of each WAV file, we assume that the pressure due to the ship at each moment in time is not correlated with the pressure due to other (background) noise sources. Thus, the power received from the ship is the average total power minus the average background power: (2) PwrS=PwrT−PwrB.

We estimate PwrB for each passing ship as the average of the power in two samples—the quietest 2-second sample from the hour before the ship is recorded, and quietest from the hour after the ship passage.

On occasion during daylight hours, ship recordings contain noise from vessels unequipped with AIS (usually recreational motorboats and occasionally larger vessels operating without AIS). This contamination is limited to the 50, 75, and 95% quantiles above 20,000 Hz, has peak spectrum levels near 50,000 Hz—a frequency commonly used for depth sounders—and is rare, but we have nevertheless reduced it via a 2-step statistical process.

Since it is very rare to have motorboat noise overlapping with ship passage at night, we first determined the 95% quantile of each received spectrum level across all vessels recorded at night (hour of day greater than 19:00 or less than 07:00) and used it as a threshold above which contamination by boat noise may have occurred. Then we re-processed all ship transits, removing any data points for which the threshold was exceeded. Any recording in which at least 100 of the 8,192 spectral received levels were above threshold was omitted from further statistical analysis.

Through this robust statistics process, about 15% of transits were omitted, resulting in no difference between the ship population quantiles for ships that pass during the day versus the night. A sensitivity analysis shows that the process did not affect the 5%–75% quantiles and that the 95% quantile was reduced by less than 2 dB—and only above about 20,000 Hz. The high frequency peaks seen in the 95% quantile in Fig. 3 become sharper as the threshold is increased or the total number of vessels analyzed is decreased.

Figure 3 Quantiles of received spectra for background and ship noise.

Quantiles (5, 25, 50, 75, & 95%) of background spectrum level (RLB, dashed blue lines) and total received spectrum level for the entire ship population (RLT, solid black lines).

Finally, we report received levels (RL) in decibels relative to a reference pressure of 1 µPascal and estimate ship received levels as: (3) RLS=10log1010RLT∕10−10RLB∕10.

Often RLT is much higher than RLB at all frequencies. In such cases, subtraction of the background has little effect on RLS. But for many ships RLT is close to RLB, at least at some frequencies. Therefore, we subtract the estimated background from the RLT at all frequencies for every isolated ship, yielding the received spectrum level of ship noise, RLS.

We cannot determine RLS if the associated RLT is not greater than RLB. Hence we require that RLT at any given frequency must exceed a threshold of three times the background spectrum level at that frequency. We choose this factor (4.8 dB) by examining the statistics of typical ship and background recordings to assure that noise is unlikely to be taken as signal. We refrain from reporting ship source spectra above 40,000 Hz because the sample size in bands above this frequency falls below about 10% of the mean sample size in lower frequency bands. Furthermore, to calculate broadband source levels with or without absorption we integrate the spectrum levels only up to this 40,000 Hz upper limit.

Prior to the background subtraction, our data commonly contained narrow-band noise peaks near 25, 38, 43, 50 and 69 kHz in many of the background and total received spectrum level quantiles (Fig. 3). Unknown sources of transient systematic noise (most commonly near 77 kHz), typically lasted only a few days. Because these noise sources are narrow or brief, they contain little power. Also, since they occur in both the received level and background data, they tend to be removed through background subtraction, and therefore do not significantly contaminate the estimated source levels (Fig. 4). One exception is the peak near 25 kHz—likely associated with the Jim Creek Naval Radio Station (transmitting at 24.8 kHz)—which persists in many source level spectra, probably indicating that the submarine communications are intermittent, at times occurring during a ship passage but not during the corresponding background measurements.

Figure 4 Ship noise source spectrum, 1/12-, and 1/3-octave levels.

Source level (SL) spectra of the entire ship population in 1 Hz (solid), 1/12-octave (dashed), and 1/3-octave bands (dotted). Black curves are medians without absorption; red curves are medians with absorption. For the spectrum levels, we delineate 25 and 75% quantiles in lighter tones. Levels with absorption start to increase rapidly above 15,000–20,000 Hz for both the 1/12- and 1/3-octave bands.

Transmission loss experiment

To estimate the source spectrum level of isolated ships from RLS we measured the transmission loss along the 240° true bearing line from the near-shore hydrophone at Lime Kiln into the northbound shipping lane (Fig. 1). The transmission loss is a combination of geometric spreading and frequency-dependent absorption. While Haro Strait has less distinct winter and summer sound speed profiles than other parts of the Salish Sea due to vertical mixing by tidal flow over bounding sills, to average out any seasonal effects we conducted our transmission loss experiment in the spring.

We determined the geometric spreading via a field experiment conducted in March, 2014, from a 10 m catamaran. We projected a sequence of 2-second tones (Table 1) using a Lubell 9816 underwater speaker lowered in a bifilar harness from the bows and attached to a power amplifier and a digital sound player. During each tone sequence, we noted the location of the projector on the sailboat’s GPS and measured the projected sound level with the Interoceans 402 hydrophone, having positioned its calibrated hydrophone near the stern, about 10 m from the projector. We oriented the projection system toward the lighthouse as we played each sequence at the following distances from the projector to the Lime Kiln hydrophone: 290; 1,035; 1,446; and 2,893 m.

Table 1 Results of the transmission loss experiment.

For each projected frequency, the geometric spreading rate (TL) is near-spherical, with an average slope of −18.6 ± 0.4 dB/decade.

Frequency (Hz)	TL (dB/decade)	Coefficient of determination	
630	−18.85	0.926	
1,260	−18.08	0.991	
2,510	−18.99	0.986	
5,000	−18.24	0.964	
10,000	−18.37	0.974	
15,000	−19.09	0.987	
20,100	−18.67	0.971	

This study focuses on determining the source levels of ships that are northbound at Lime Kiln lighthouse. By limiting our analysis to northbound vessels we reduce the difficulty of determining accurate transmission loss by limiting the variation in range of the targets. Furthermore, our underwater speaker used to measure transmission loss did not have sufficient power especially at high frequencies (near 20,000 Hz) to provide detectable signals at ranges much larger than the 2,893 m range that brackets the more distant edge of the north bound traffic lane.

We analyzed the spreading of the test tones by measuring the calibrated RMS level received at the Lime Kiln hydrophone for each tone at each distance. The received signal was determined by subtracting the calibrated background level from the received level of the corresponding tone (Eq. (3)). To determine the geometric spreading contribution to transmission loss, we added to the received signal levels the amount of absorption expected for each frequency and range (straight line path, R). Following Francois & Garrison (1982) we used R to calculate the absorption loss at each frequency. For our highest test tone frequencies and range, accounting for absorption added from 2 dB (at 10,000 Hz) to 8.6 dB (at 20,000 Hz) back into the received signal levels.

We used linear regression to model the absorption-corrected received signal levels as a function of the base 10 logarithm of the range from receiver to source in meters separately for each of our test tones. The slopes and goodness of fit are shown in Table 1. Since these slopes are not correlated with the frequency (correlation coefficient of 0.003), we average them and use the resulting near-spherical geometric spreading coefficient (transmission loss coefficient, TL) of −18.6 ± 0.4 dB/decade in log10(R) to represent geometric spreading out to a distance of about 3 km. Also, as these slopes vary little over a factor of 30 in frequency, we assume that we can use this mean slope to extrapolate down from 630 Hz to our 20 Hz lower frequency cutoff and up from 20,000 Hz to our 96,000 Hz upper frequency Nyquist cutoff.

Source levels

We calculate source spectrum levels of ship noise without absorption (SL) in Eq. (4) and then with absorption (SLa) in Eq. (5), determining α from Francois & Garrison (1982). (4) SL=RLS+18.6log10R

(5) SLa=RLS+18.6log10R+αfR.

We integrate the source spectrum levels from 11.5 Hz up to 40,000 Hz to compute broadband source levels (SL) (Table 2). We also integrate the source spectrum levels over both 1/3-octave and 1/12-octave bands with band centers determined by fi=fo2iN where i is an integer and N is the number of partitions of each octave. This is both consistent with ISO center frequencies (ISO 266) and allows comparison with the proposed annual mean noise thresholds at 63 and 125 Hz Tasker et al. (2010); Merchant et al. (2014). Finally, when plotting quantiles of levels we exclude the lowest frequency bin (11.5 Hz) because for some classes an insufficient number of ships passed the 4.8 dB re 1 µPa signal-noise threshold to estimate the 5% and 95% quantiles.

Table 2 Ship population statistics and mean broadband sound pressure levels (20–40,000 Hz).

Though abbreviated in the table as dB, the units of the received signal levels (RLS) are dB re 1 µPa and source levels have units of dB re 1 µPa @ 1 m. Variability is reported as a standard deviation of the mean, and speed over ground (SOG) is provided in m/s and knots.

Ship class	Isolated transits	% of total	Unique ships	RLS dB	SL dB	SOG m/s	SOG knots	
All classes combined	2,809		1,582	110 ± 7	173 ± 7	7.3 ± 2.0	14.1 ± 3.9	
Bulk carrier	965	34.3	734	111 ± 6	173 ± 5	7.0 ± 0.8	13.7 ± 1.5	
Container	529	18.8	207	116 ± 4	178 ± 4	9.9 ± 1.0	19.2 ± 1.9	
Tug	337	12.0	85	108 ± 5	170 ± 5	4.2 ± 1.2	8.2 ± 2.3	
Cargo	307	10.9	206	113 ± 5	175 ± 5	7.4 ± 1.2	14.4 ± 2.3	
Vehicle carrier	187	6.6	111	113 ± 3	176 ± 3	8.7 ± 1.0	16.9 ± 1.8	
Tanker	148	5.3	101	111 ± 5	174 ± 4	7.1 ± 0.7	13.8 ± 1.4	
Military	113	4.0	19	99 ± 10	161 ± 10	5.7 ± 1.6	11.1 ± 3.1	
Fishing	65	2.3	28	102 ± 9	164 ± 9	4.7 ± 1.1	9.1 ± 2.2	
Passenger	49	1.7	31	104 ± 8	166 ± 8	7.4 ± 2.3	14.4 ± 4.5	
Miscellaneous	41	1.4	21	101 ± 9	163 ± 9	5.8 ± 3.0	11.2 ± 5.8	
Pleasure craft	41	1.5	35	97 ± 10	159 ± 9	6.4 ± 2.5	12.4 ± 4.9	
Research	14	0.5	4	104 ± 6	167 ± 5	5.7 ± 0.9	11.1 ± 1.8	

To facilitate comparison with past studies we generally present ship source spectrum levels as SL. However, due to the presence of high-frequency ship noise in our recordings and its potential impact on marine life exposed at close range, we also present absorption-corrected spectral power levels (SLa) for the whole ship population.

Results and Discussion

Ship traffic patterns

Combining all ship classes over the entire study, our data set describes 1,582 unique vessels that made a total of 2,809 isolated, northbound transits of the shipping lanes in Haro Strait (Table 2). The 2,809 isolated transits sample 17.1% of the total transits through Haro Strait (16,357, northbound and southbound) logged by our AIS system during the study period. Of 7,671 total northbound transits, 36% were sampled, suggesting that about 2/3 of the traffic in Haro Strait is not isolated. Dividing the total transits by the 850 day study period shows that the average daily ship traffic is 19.5 ships/day. This amount of traffic is comparable to previous estimates for Haro Strait: about 20 ships/day (Veirs & Veirs, 2006) and about 1 ship/hour (Erbe, MacGillivray & Williams, 2012).

About 1/3 of the isolated transits are bulk carriers and about 1/5 are container ships. The next 4 most prevalent ship classes—tugs, cargo ships, vehicle carriers, and tankers—constitute another 1/3 of the isolated transits. Of the remaining less-prevalent ship classes, we sample military ships 113 times (19 unique vessels), and other ship classes 18–65 times.

Together, bulk carriers and container ships comprise more than half (53%) of the isolated shipping traffic in Haro Strait. About 3/4 of isolated bulk carrier transits are unique vessels, in contrast to container ships which are unique only about 40% of the time. This may indicate that the global bulk carrier fleet is larger than the container fleet, or that shipping economics or logistics limit the diversity of container ships transiting Haro Strait. For example, container ships may ply routes that are more fixed, and therefore repeat transits through Haro Strait more frequently than bulk carriers.

Those ship classes that have many isolated transits by a small number of unique ships offer us opportunities to study variability of noise from individual ships. Military vessels, a category with 19 unique ships sampled on 113 isolated transits, have about 7 isolated transits per unique ship, while tugs and research vessels have about 4 and container ships have about 3.

Broadband levels

Received levels

Broadband population mean received levels (RLS, Table 2) vary between ship classes from a low of 97 dB (pleasure craft) to a high of 116 dB (container ships). Combining all classes, RLS is 110 ± 7 dB which is 19 dB re 1 µPa above the mean background level (RLB) of 91 ± 4 dB. These levels are comparable to anthropogenic and background received levels noted in previous studies at similar distances to shipping lanes and over similar frequency ranges (Veirs & Veirs, 2006; McKenna et al., 2012). While our RLS from ships 0.95–3.65 km away is 10–22 dB lower than the 121–133 dB reported by Bassett et al. (2012), only about 2 dB of this difference can be explained by the shorter distances to their ships (0.58–2.82 km).

Source levels (SL)

The mean broadband source level (SL, Table 2) for all ship classes combined is 173 ± 7 dB re 1 µPa @ 1 m. Comparing between ship classes, container ships have the highest SL at 178 dB. Other classes with SL ≥ 173 dB include vehicle carriers, cargo ships, tankers, and bulk carriers. Tugs, research, and passenger vessels (primarily cruise ships, as there are no nearby ferry routes) have SL of 166–170 dB, while the remaining vessel classes have SL from 159–164 dB. This range of SL across classes (159–178 dB) overlaps the 170–180 dB range specified for small ships (lengths 55–85 m) by Richardson et al. (1995). When frequency dependent absorption is included, mean broadband source levels increase by 0.5–1 dB (we have limited the upper frequency to 40,000 Hz).

Our range of mean values is similar to recent estimates of broadband source levels for similar-sized modern vessels, but for some classes other estimates are 1–11 dB higher than our estimates. Figure 2 depicts broadband SL statistics for each class we studied and juxtaposes the results from other studies of modern ships for comparable classes. Some of these studies are discussed below, partially to assess our results and partially to consider some of the common ways in which methods may differ between studies of ship noise: sample sizes, bandwidths, averaging times, calibration procedures, background subtractions, absorption or other frequency-dependent corrections, geometric spreading rates, and ship characterization (e.g., classification, criteria for isolation, speed, and range).

Compared with mean broadband source levels (20–30,000 Hz, TL of −15, absorption assumed negligible) computed by Bassett et al. (2012) our means are 0–6 dB lower, depending on the class. The comparatively low values of our means cannot be explained by distinct methodology; their study used a narrower broadband bandwidth and a lower (modeled) transmission loss. The most likely explanation for the differences in most classes is a difference in distinct ship design and/or operating characteristics between Puget Sound and Haro Strait populations. There is some evidence that ships measured by Bassett et al. (2012) may have higher speeds than in our study. Of the 24 select ships for which Bassett et al. (2012) provide speed data, 38% have SOG greater than 1 standard deviation above our mean values for the corresponding class. The average elevation of SOG for those ships is +2.0 m/s (+3.8 knots).

Compared with broadband source levels (20–1,000 Hz, TL of −20) listed for 29 individual ships by McKenna et al. (2012) the mean values for equivalent classes in Table are 1–13 dB lower. These differences are also depicted in Fig. 2. Accounting for the difference in TL (1.4 dB/decade of range) between the studies would raise our SL values an average of 4.7 dB, thereby causing our inter-quartile range to overlap with or encompass the ranges of McKenna et al. (2012) for all comparable classes except bulk carriers. As with the Bassett et al. (2012) study, adjusting for differences in broadband bandwidth would raise their individual ship source levels even higher above our means, so cannot help explain the differences. Examining the SOG differences by class offers less of an explanation in this case; of the 29 ships, only 3 (about 10%) have speeds that exceed our mean SOG in the associated class, and only by an average of 1 m/s (2 knots).

A study of 593 container ship transits by McKenna, Wiggins & Hildebrand (2013) yielded a mean source level (20–1,000 Hz, TL of −20) for the population of 185 dB, 7 dB higher than our mean of 178 dB for 529 container ship transits. In Supplemental Information 1, McKenna, Wiggins & Hildebrand (2013) provide a mean speed of 10.5 ± 1.4 m/s—roughly 0.5 m/s above our container ship mean speed of 9.9 m/s—and an mean range of 3,246 ± 291 m (about 1 km larger than our mean range). The speed difference could only account for about 0.5 dB of the source level discrepancy between the studies, based on the +2.2 dB per m/s (+1.1 dB/knot) relationship between broadband source level and speed portrayed for a single ship in McKenna, Wiggins & Hildebrand (2013).

Compared with broadband source levels (45–7,070 Hz) of individual vessels measured by Malme, Miles & McElroy (1982) and Malme et al. (1989) and tabulated by Richardson et al. (1995) our means for respective classes are 1 dB lower than a tug (171 dB at 5.0 m/s (9.7 knots)), 6 dB lower than a cargo ship (181 dB), and 12 dB lower than a large tanker (186 dB). These differences might be due to more modern ships decreasing their speed (at least while in coastal waters) or increasing their propulsion efficiency.

Kipple (2002) measured 6 cruise ships at a range of 500 yards and reported broadband source levels (10–40,000 Hz, TL of −20, absorption ignored) of 175–185 dB re 1 µPa 1 yard at 10 knots (5 m/s) and 178–195 dB re 1 µPa @ 1 yard at 14–19 knots (7–10 m/s). In comparison, our population of passenger ships (including cruise ships) has a mean SL of 166 ± 8 dB re 1 µPa @ 1 m and a mean speed of 14.4 ± 4.5 knots. Thus, while the speeds tested by Kipple (2002) bracket our mean speed, our mean SL is 9–29 dB lower than their range of source levels. One possible explanation for this difference is an unspecified upward correction of received levels below 300 Hz that Kipple (2002) made to account for multipath propagation effects. This is substantiated by Malme et al. (1989), who state that passenger vessels in Southeast Alaska have SL from 170 to180 dB, a range that falls between our mean and maximum SL for passenger vessels and mostly below the ranges given by Kipple (2002).

Finally, Arveson & Vendittis (2000) measured a bulk carrier at 8–16 knots (4–8 m/s) and found broadband source levels (3–40,000 Hz, TL of −20) of 178–192 dB. The source levels they calculated for speeds of 12 knots (6 m/s) and 14 knots (7 m/s), 184 dB and 190 dB, respectively, are most comparable to our bulk carrier population with SOG of 7.0 ± 0.7 m/s. Without correction for the different transmission loss assumptions, our bulk carrier SL of 173 ± 5 dB is 11–17 dB below their levels.

While this pattern could be interpreted as an underestimation of SL by our methods, we believe our population statistics represent an accurate estimate of source levels for modern ships operating in coastal waterways. In almost all of the cases that we have discussed, the maximum discrepancy is less than 1.5 times the inter-quartile distance (25% vs 75% quantiles) for the comparable ship class (see Fig. 2). Exceptions are some of the louder container ships in McKenna, Wiggins & Hildebrand (2013) and vehicle carriers in McKenna et al. (2012), the large tanker mentioned in Richardson et al. (1995), the higher-speed cruise ships of Kipple (2002), and the bulk carrier of Arveson & Vendittis (2000) when its speed was greater than 8 knots (4 m/s).

Even these exceptional upper values from the literature are almost completely contained within the distribution of our broadband SL population. Our maximum SL for a bulk carrier (191 dB) is 3.6 dB higher than the loudest bulk carrier tabulated in McKenna et al. (2012) and above the bulk carrier source levels obtained by Arveson & Vendittis (2000) at all speeds except 16 knots, or 8.2 m/s (192 dB). The loudest bulk carrier tabulated in Bassett et al. (2012) with source level of 182 dB is equal to the 95% quantile of SL within our bulk carrier class. The loudest ship tabulated by Richardson et al. (1995), a tanker with SL of 186 dB, is only 0.8 dB above our loudest tanker. One explanation for this outlier is that the ship was a supertanker driven by a steam-turbine—and therefore may represent the “upper range of large merchant vessels” (Malme et al., 1989). Finally, our passenger vessel population has a 95% quantile of 177 dB and a maximum of 183 dB, a range that encompasses most of the slow ships and the lower portion of the faster ships assessed by Kipple (2002).

Across all classes, the maximum broadband SL for an individual ship was 195 dB for a container ship, 7 dB above the highest overall values reported by McKenna et al. (2012) and Bassett et al. (2012)— both for container ships, as well. Our maximum is consistent with the study of 593 container ships by McKenna, Wiggins & Hildebrand (2013) in which the maximum source level was also 195 dB. Our second- and third-highest maxima within a class were from a bulk carrier (191 dB) and a cargo ship (186 dB). All other classes had maximum SL ≤ 185 dB m. The lowest maximum SL within a class was 176 dB for pleasure craft.

The range of minimum broadband SL across all classes in our study was from 130 dB for a cargo ship to 167 dB for a vehicle carrier. In comparison, McKenna et al. (2012) reported a minimum SL across all classes of 177 dB for a chemical tanker, while the minimum SL for a container ship in McKenna, Wiggins & Hildebrand (2013) was 176 dB. In contrast with the exact agreement of the maxima between our container ship population and the data set of McKenna, Wiggins & Hildebrand (2013), this discrepancy of at least 9–10 dB in SL minima suggests that methodological differences between the studies may exert greater bias when ship signal levels are near background noise levels.

Ship speed

Averaged across all vessels, the SOG of isolated ships northbound in Haro Strait is 7.3 ± 2.0 m/s (14.4 ± 3.9 knots). This is higher than the mean of 10–12 knots (5.1–6.2 m/s) observed during WWII, but possibly lower than the post-war (mid-1970s) mean of about 15 knots (7.7 m/s) (Ross, 1976).

In our study, the fastest classes are container ships (mean SOG of 9.9 m/s) and vehicle carriers (8.7 m/s), while the slowest vessels are fishing boats (4.7 m/s) and tugs (4.2 m/s). For tankers, our SOG of 7.1 ± 0.7 m/s is slightly below the 7.2–8.2 m/s (14–16 knot) range reported by Ross (1976) for both “T2 tankers” in WWII and supertankers built after about 1960.

Overall, our data set samples a small range of ship speeds within any given class. Because Haro Strait is relatively long and straight, most vessels transit it without changing speed. Whether north- or south-bound, they have consistent SOG means and standard deviations. This low variability in speed limits our ability to search for relationships between noise and speed, but may help us discern in future work the influence of other variables—like propeller type, draft (loading), or maintenance levels—building on insights from McKenna, Wiggins & Hildebrand (2013).

Relationship between speed and broadband source level

Upon linear regression of SL versus SOG for all data, we find a slope of +1.8 dB per m/s (+0.93 dB/knot). The coefficient of determination (R2) for this fit explains only 27% of the variance in the data (assuming normal distribution). Furthermore, most of the variation in SL is likely driven by ship class (which was not controlled for in the regression), with little change in speed within ship class. Slopes vary from +0.2 to +3.4 dB per m/s between ship classes. Examination of repeated transits of individual ships shows that the variation in slope is high between individual ships within a class and the percent of variance explained is low. While slopes are positive for most individual ships, some are zero or negative. These variations indicate that the overall population slope should not necessarily be applied to all ship classes or individual ships, echoing the recommendations of McKenna et al. (2012).

Received spectra

Most ships transiting Haro Strait raise background noise levels in the core summertime habitat of SRKWs at all measured frequencies (Fig. 3). Specifically, 95% of the ships generate received spectrum levels at or above the 95% quantile of background levels from 20–96,000 Hz. Thus, at ranges of a couple kilometers, commercial ships cause significant underwater noise pollution not only at low frequencies, but also at high-frequencies.

The difference in median spectrum levels between ship and background noise levels is more than 30 dB below 100 Hz and gradually decreases to about 10 dB at 20,000 Hz. In the high frequency range of 20,000–96,000 Hz the median ship noise is elevated above median background spectrum levels by at least 5 dB. This significant elevation of background levels at high frequencies is what motivated us to account for absorption when computing ship source levels and is consistent with an observation by Hildebrand et al. (2006) of a single commercial ship in Haro Strait at a range of 442 m that elevated the ambient noise spectrum levels by as much as 30–40 dB across a broad band of the spectrum (60–75,000 Hz).

If we define the 5% quantile of background noise as an “ancient” ambient condition (Clark et al., 2009) then the typical (median) modern ship raises spectrum noise levels above ancient levels by 12–17 dB at frequencies used in killer whale echolocation (20,000–70,000 Hz) and by 17–35 dB at frequencies used in killer whale social vocalization (200–20,000 Hz). In the frequency range used by vocalizing baleen whales (20–200 Hz), the median ship spectrum noise levels are about 32–35 dB above the ancient ambient levels.

We gain additional confidence in the accuracy of our sound pressure levels (and implicitly our system calibration) by comparing the received spectrum levels in Fig. 3 with ambient noise spectra from other studies. Our background quantiles are bracketed by the average deep-water ambient noise levels associated with sea state 1–3, though the slope of our median curve from 1,000–10,000 Hz is −8 dB/decade, about half as steep as the open-ocean slope of −17 dB/decade Urick (1983). The “usual lowest ocean noise” curve of Cato depicted in Plate 5 of National Research Council et al., (2003) is bounded by our 5% and 25% quantiles from about 30 to 10,000 Hz. Two ambient noise spectra obtained in Haro Strait by Hildebrand et al. (2006) have levels that are bounded by our 5% and 95% quantiles of background noise from 300 Hz to 30,000 Hz. The single ship spectrum (60 Hz–75,000 Hz) obtained opportunistically by Hildebrand et al. (2006) at a range of 442 m has levels that are greater than our 75% quantile of RLB at all frequencies.

Similarly, our quantiles of total received spectrum level are consistent with previous studies. For example, the noise spectrum levels recorded in US bays and harbors during World War II by Urick (1983) are entirely bounded by our quantiles of RLT from 100 Hz to 10,000 Hz. The peak levels (at about 50 Hz) of the shipping contribution to deep water ambient noise estimated by Ross (1976) for “remote, light, moderate, and heavy” traffic are approximately 71, 77, 85, and 95 dB, respectively; the upper three traffic levels are encompassed by our 5% and 95% quantiles, while the “remote” levels are no more than 2 dB below our 5% quantile. Finally, the quantiles of unweighted received spectrum levels in Bassett et al. (2012) peak near 50 Hz and have levels that are within about 5 dB of our levels for corresponding quantiles at all frequencies common to the two studies. Even at high-frequencies our data are consistent; Knudsen, Alford & Emling (1948) reported total received levels of 40–50 dB at 30,000 Hz in coastal waters, a range which brackets our quantiles at that frequency.

Source spectra

Median source spectra for the whole ship population are shown in Fig. 4 as spectrum, 1/12-octave, and 1/3-octave levels, with and without accounting for absorption. For the spectrum levels, we also present 25% and 75% quantiles.

Source spectrum levels without absorption

The median spectrum levels peak near 50 Hz at about 154 dB and decrease at higher frequencies with a slope of about −15 dB per decade (from 50–40,000 Hz). The 25% and 75% quantiles are 3–5 dB from the median below about 10,000 Hz, but at higher frequencies the difference decreases to about 1 dB re. In the region between 700 and 40,000 Hz the median spectrum has a subtle slope break near 5,000 Hz, with a slope of about −10 below and about −20 above.

Previous observations, models, and experimental results all help contextualize these whole-population spectrum levels. Unfortunately, many previous studies of ship noise are not comparable due to presenting species-specific band levels (e.g., Hatch et al., 2012) or band levels rather than spectrum levels, or other limitations: small sample size, non-overlapping frequency ranges, and ship classes with low diversity, distinct definitions, or incomparable ships (e.g., ice breakers in Erbe & Farmer, 2000).

One exception that allows comparison up to 1,200 Hz is the analysis of 54 ships at ranges of 360–1,800 m by Wales & Heitmeyer (2002). Their measured average source spectrum levels are bounded by our 25% and 75% quantiles from 400 to 1,200 Hz. At lower frequencies (below 400 Hz) their mean levels exceed our 75% quantile by 2–20 dB (20 dB at 20 Hz; 5 dB at 50 Hz; and 2 dB at 100 Hz). Interestingly, their curve does not peak near 50 Hz, but instead continues rising as the frequency decreases to 30 Hz, the lowest frequency they measured. The slope of their mean curve is about −30 dB/decade below 100 Hz, and −20 dB/decade above. They note that the variance around their mean levels decreases with rising frequency from a standard deviation as high as 5.32 dB below 400 Hz to about 3.12 dB above it. This suggests that a partial explanation for the elevation of their mean values relative to our 75% quantile may be variability in low-frequency power between ships.

Models of ship noise that output spectrum levels provide another point of comparison. Our 50% and 75% quantiles are encompassed in the spectrum levels presented by National Research Council et al., (2003) for 3 classes of tankers, as well as merchant and fishing classes, based on the RANDI model (Wagstaff, 1973; Breeding et al., 1994) parameterized with data from Emery, Bradley & Hall (2001) and Mazzuca (2001). The 25% quantile is also encompassed, except below 30 Hz. Below 300 Hz, our median values lie between the fishing and merchant class levels of National Research Council et al., (2003); at higher frequencies—up to 1,000 Hz, the upper limit of their estimates—our median values are above their merchant class but below their intermediate tanker class (length 153–214 m, speed 7.7–9.3 m/s). Overall, this comparison suggests that our median source level spectra validate the RANDI model as parameterized in National Research Council et al., (2003) at intermediate frequencies (100–1,000 Hz), but below 100 Hz our median levels are lower (by about 5–30 dB) than the RANDI model predicts for all classes except fishing vessels (length and speed bins of 15–46 m, 3.6–5.1 m/s).

Other noticeable differences between our population median spectrum levels and those modeled in National Research Council et al., (2003) are the frequency of the peak power, the general slope of the spectra above the peak, and secondary peaks resolved in our data. While our spectra peak near 50 Hz, the peak power in the spectra of National Research Council et al., (2003) occurs slightly lower, at 30 Hz. Between 100 and 1,000 Hz, the slope of our median spectrum is −12 dB per decade, nearly three times less steep than the slope of −35 dB per decade in National Research Council et al., (2003). Our spectrum levels have detailed structure where the RANDI model curves of National Research Council et al., (2003) are smooth. Our quantiles show secondary power peaks between 80 and 1,100 Hz and many narrowband peaks in 1,100–10,000 Hz range, similar to the frequency dependence of spectral line complexity observed by Wales & Heitmeyer (2002).

Experiments with cavitation provide a final comparison with our whole-population spectrum levels. Above 5,000 Hz the slope of our median spectrum matches the slope observed during cavitation of a spinning rod (Mellen, 1954) and a water jet (Jorgensen, 1961)—−20 dB per decade, (or −6 dB per octave).

Source spectrum levels with absorption

The spectrum levels with absorption are indistinguishable from those without absorption below about 5,000 Hz. At higher frequencies, the SLa median spectrum level curve diverges from the SL curve, and starts to rise rapidly at the 40,000 Hz cut-off of this study. The associated 25% and 75% quantiles are within 3–5 dB of the median values throughout the region of divergence.

These alternative source spectra look unfamiliar at high frequencies, and are not consistent with available data taken close (less than 500 m) to ships. For example, the single container ship measured at a range of 442 m by Hildebrand et al. (2006) in Haro Strait has a absorption-corrected source spectrum level of 108 dB re 1 µPa2∕Hz @1 m at 40,000 Hz—about 17 dB below our SLa spectrum level at that frequency.

However, we believe the absorption-corrected spectra in Fig. 4 are rooted in accurate physics and we note that the spectrum levels of SLa are in agreement with some measurements of underwater noise radiated during fully developed cavitation. For example Lesunovskii & Khokha (1968), specify rotating bar noise spectrum levels of 95–115 dB at 10,000 Hz while our 25%–75% quantiles of SLa spectrum level at that frequency are 114–120 dB. Similarly, Blake, Wolpert & Geib (1977) report noise levels from a cavitating hydrofoil of 75–110 dB re 1 µPa2∕Hz @ 1 yd at 31,500 Hz which is approaching our 25%–75% quantiles of SLa at that frequency (120–125 dB re 1 µPa2∕Hz @ 1 m).

We expect that propeller cavitation noise intensity will be greater than laboratory measurements due to increased length scale and number of the blades on ships. Evidence from World War II studies of torpedo and submarine noise attributed to cavitation supports this expectation. Figures 10.21–10.23 of Urick (1983) show levels equivalent to or bracketing our SLa spectrum levels: 24,000 Hz spectrum levels of 118 dB re 1 µPa2∕ Hz @ 1 yd for a submarine cruising at 8 knots (4 m/s) near periscope depth; 25,000 Hz spectrum levels of 100–130 dB re 1 µPa2∕Hz @ 1 yd for torpedos moving at 20–45 knots (10–23 m/s); and 20,000 Hz spectrum levels of 115–130 dB re 1 µPa2∕Hz @ 1 yd for a suite of torpedoes.

Source 1/12- and 1/3-octave levels

The median 1/12- and 1/3-octave level curves in Fig. 4 are elevated relative to the median spectrum levels and diverge from them above 50 Hz due to the integration of spectrum levels over bands that get progressively wider with increasing center frequency. Like the spectrum levels, these curves have a peak near 50 Hz. Peak values are 158 dB re 1 µPa2 per band @ 1 m for the 1/12-octave levels and 163 dB for the 1/3-octave levels. Above 50 Hz, both curves have slopes of about −4 dB/decade from 100 to 5,000 Hz, −10 dB/decade from 5,000 to 40,000 Hz.

While we are unaware of a comparable aggregation of source spectra from multiple ship classes presented as 1/3-octave levels, there are many studies of individual ships or classes that present 1/3-octave source levels. We compare them here with the median 1/3-octave curve for our ship population because we present only spectrum levels when assessing inter- and intra-class differences in subsequent sections.

Our median 1/3-octave levels are entirely bounded by the estimated levels for 6 diverse ship types presented in Figure 3.14 of Malme et al. (1989) at all comparable frequencies (20–16,000 Hz). Similarly, our levels are within the estimated 1/3-octave source levels (10–10,000 Hz) summarized in Figure 6.5 of Richardson et al. (1995) for an ice breaker, a composite of supertankers, and a tug/barge at almost all frequencies. Only above about 2,000 Hz is our median curve slightly below comparable vessels described by Richardson et al. (1995): ours is within 2 dB of their tug/barge levels, and no more than 10 dB below their supertanker levels. Overall, we find the consistency of our results with these two studies to be remarkable.

Comparing our median curve with the 7 ships (representing five of our classes) for which McKenna et al. (2012) presented 1/3-octave levels, our levels are 5–10 dB lower at all common frequencies (20–1,000 Hz). As discussed when presenting spectrum levels, we are not sure how to account for this difference, other than to recognize key differences between the studies: distinct transmission loss, our much larger sample size, and our higher diversity of classes.

Studies of ship noise in which speed was varied present a range of levels that is also consistent with our results. Compared with the maximum–minimum envelopes of 1/3-octave source levels (referenced to 1 yard) from 6 cruise ships presented by Kipple (2002) our 1/3-octave levels are within the envelope for both 10 knot (5 m/s) and 14–19 knot (7.2–9.8 m/s) samples, except below 25 Hz where our levels are lower by 1–7 dB. Our levels also fall within (but near the lower edge) of the range of 1/3-octave spectra reported by Arveson & Vendittis (2000) for a bulk carrier tested from 68 to 148 rpm.

Our 1/3-octave levels help validate the RANDI model used by Erbe, MacGillivray & Williams (2012) to compute 1/3-octave spectra for five ship length classes over a range of speeds observed in traffic off the coasts of British Columbia and Washington State. Overall, our median levels are entirely within the range of their estimated levels at all modeled frequencies (10–2,000 Hz). More specifically, though, our median crosses their size-specific curves, because it has a less steep slope. Below 400 Hz our levels are bounded by their L1 and L3 classes (representing lengths less than 50 m); above 400 Hz our median levels are between their L4 and L5 classes (greater than 50 m).

An even more dramatic crossing of model curves by our median 1/3 octave source spectrum level curve is evident upon comparison with Figure 1 of Williams et al. (2014). While our median source levels are equivalent to or bounded by the 1/3-octave levels for each of their modeled ship types (tug, cruise ship, container ship) near or below 250 Hz, at higher frequencies our levels exceed the modeled ones by 7–10 dB.

The crossing of such modeled spectra by our 1/3-octave median curve is one manifestation of a subtle slope difference between our results and previous studies (Arveson & Vendittis, 2000; Kipple, 2002; Erbe, MacGillivray & Williams, 2012; Williams et al., 2014). While Arveson & Vendittis (2000) observe slopes from above a 55 Hz cavitation “hump” up to about 30,000 Hz to be −10 dB/decade on a 1/3-octave plot, our slope over the same frequency range is shallower (−6.5 dB/decade) and we observe a slope break near 3,000 Hz. Below the break the slope is about −4.5 dB/decade, while above it is −10 dB/decade.

The similarity of our 1/3-octave levels with those from available studies at frequencies below 630 Hz (the lowest tone used in our transmission loss experiment) is the first evidence that our measurements of low-frequency radiated noise are accurate. The lower slope relative to other studies suggests that the ship population in this study is generating proportionally more high-frequency noise than ships in previous studies.

Source 1/12- and 1/3-octave levels with absorption

As with the spectrum levels, the 1/12- and 1/3-octave level curves with absorption are indistinguishable from those without absorption below 5,000 Hz. At higher frequencies, the SLa median 1/12- and 1/3-octave levels rise to match the 50 Hz levels of the associated median SL curves near 35,000 Hz and then continue to increase at higher frequencies.

This means that when we account for absorption when computing 1/12- or 1/3-octave levels, modern ships radiate noise in high-frequency bands (centered near 35,000 Hz) at levels equivalent to the low-frequency maxima near 50 Hz. This surprising equivalency, and the theoretically even higher power levels in bands above 35,000 Hz, are important to consider when assessing the masking potential of ship noise in habitats close to or within shipping lanes for marine species that utilize high-frequency signals. Although it is novel to state that ship noise source levels have peak power at high- as well as low-frequencies, we provide these 1/12- and 1/3-octave noise levels to facilitate accurate modeling of acoustic impacts for species that have critical bands overlapping these octave bands (Richardson et al., 1995).

While the median 1/12-octave source levels reported by Erbe & Farmer (2000) for the cavitating propeller of an ice breaker are not comparable to any of our ship classes (and much higher—30 dB re 1 µPa2 per band @ 1 m above our median level at their power peak near 500 Hz), we note that the slope of their median curve is −13 dB/decade from 1,000 to 10,000 Hz. Importantly, Erbe & Farmer (2000) is rare in stipulating that absorption was accounted for in computing source levels. Their slope is about twice as steep as our 1/12-octave median slope of −7 dB/decade in the same frequency range.

Finally, Kipple (2002) did not correct for absorption, but made measurements of cruise ship receive levels up to 40,000 Hz at short range (500 yards) and therefore provides a rare point of reference for our high-frequency SLa levels. Their 1/3-octave band source levels at 40,000 Hz for a suite of cruise ships and speeds (10–19 knots; 5.1–9.7 m/s) vary from 133 to 154 dB re 1µPa @ 1 yard—values that approximately bracket our uncorrected SL levels but are 13–34 dB below our SLa levels.

Spectral differences between ship classes

When the ship population is broken down by class (Fig. 5) the medians show a striking bifurcation. While all classes have similar median spectrum levels near 20,000 Hz, the curves diverge at lower frequencies, and below 200 Hz they bifurcate into high- and low-power groups. The high-power group has peak power of 153–159 dB near 50 Hz (just above the population median shown in Fig. 4) and consists of container ships, vehicle carriers, cargo ships, bulk carriers, and tankers. The low-power group has peak power of 134–141 dB near 50 Hz or just above 100 Hz—levels well below the population median or even 25% quantile—and consists of passenger vessels, tugs, military, research, fishing, miscellaneous, and pleasure vessels.

Figure 5 Median source spectra of ship noise for different classes of ships.

Comparison of median source spectrum levels (without absorption) between ship classes.

The 25%, median, and 75% spectrum levels at the power peak near 50 Hz in Fig. 4 bracket the 50 Hz levels of the high-power group of ships in Fig. 5. The median of the whole population is most similar to the spectra in the high-power group (e.g., the bulk carrier curve) because the aggregated sample size is much higher in the high-power group than in the low-power one (see Table 2). Modelers interested in assessing impacts of specific ship classes, particularly those in the lower-power group, should not use the median or 25% quantile levels for the whole population, but instead select class-specific levels from the curves in Fig. 5.

Container ships have the highest median source spectrum level of all classes at almost all frequencies below 10,000 Hz with peak power of 159 dB near 40 Hz. This is likely because of their relatively large size and high mean speed (10 m/s) compared to pleasure craft or military ships—the classes with the lowest median power at all frequencies below 400 Hz.

Many of the ship classes show secondary peaks in the median spectrum levels from 100 to 5,000 Hz. For example, most classes show a 2 dB dip near 250 Hz and at least container ships, vehicle carriers, cargo ships, and tankers have peaks near 300, 700, and 1,000 Hz. There are also narrower peaks for these same classes between 1,000 and 10,000 Hz, most prominently at 2,000 Hz and near 3,000 Hz.

The variability of the median source level in each class decreases above 5,000 Hz and remains low until about 10,000 Hz. At higher frequencies (10,000–40,000 Hz) the variability increases again for most ship classes, but the degree of increase is a strong function of sample size within a class. While we know from examining spectrograms from individual ships that some of the narrow peaks are associated with active acoustic sources (depth sounders, scientific echosounders, and fish finders), in Fig. 5 the high variance above 10,000 Hz is due primarily to some ships having spectrum levels that do not meet the robust threshold at higher frequencies. Particularly in classes where the sample size is already small this leads to some high frequency bins having many fewer data points than adjacent bins which in turn results in more-variable median values across this high-frequency range.

The quantiles of source spectrum level by class in Fig. 6 provide further detail about inter-class differences. Comparing the 95% quantiles, container ships still have the highest peak power (165 dB) near 50 Hz, but bulk and vehicle carriers, cargo ships and tankers also have peak power greater than 160 dB. Other classes have peak power in the 95% quantiles near 50 Hz at spectrum levels that range from 156 dB (research) to 150 dB (tugs). Comparing the 5% quantiles, we expected that the military class would have the lowest levels due to more advanced ship-quieting technologies. While the military class levels are much lower than container ships (10 dB less at 1,000 Hz and 20 dB less at 100 Hz), other classes have even lower levels at those frequencies, particularly fishing vessels and pleasure craft.

Figure 6 Quantiles of ship source spectra for different classes of ships.

Quantiles of source spectrum levels for each class of ship. Median (50%) quantile (black) overlies 5, 25, 75, and 95% quantiles (blue).

Spectral variability within ship classes

All classes of ships have spectrum levels that vary more at low frequencies than at high frequencies (Fig. 6). Near 50 Hz there is a 15–35 dB difference between the 5% and 95% quantile levels. That difference decreases with rising frequency until above 20,000 Hz it is typically less than 10 dB.

Below 20,000 Hz, source level variability in Fig. 6 tends to be lower for the classes that have smaller speed over ground standard deviations and that have larger sample size as shown in Table 2. While container and cargo ships, bulk and vehicle carriers, and tankers have 95–5% spectrum level differences of about 15 dB, the other classes exhibit larger differences up to 25–30 dB. The classes with the largest number of vessels are most uniform in their speed over ground and most consistent in their vessel design and operation. Tugs are a special case because there are many transits and their speed is not unusually variable, but their loading is. Our passenger vessels are all cruise ships and hence similar in design, but their speeds are quite variable as they adjust their arrival times in the Port of Vancouver. Finally, the small numbers of pleasure craft and vessels classed as miscellaneous are highly variable in both their designs and their operations.

Other studies have observed a similar pattern of source level variability with frequency. In mean source spectrum levels from 54 ships Wales & Heitmeyer (2002) noted higher, more-variable standard deviations from 30 to 400 Hz and lower, more-constant ones from 400 to 1,200 Hz. Figure 8 of McKenna, Wiggins & Hildebrand (2013) displays histograms of octave-band power for 593 container ships which have widths that decrease from about 35 dB in the 16 Hz band to 26 dB in the 500 Hz band.

One explanation for this pattern is that the low-frequency portion of ship noise spectra is influenced by diverse design and operational details (many sources of variability), while cavitation generates high-frequency broadband noise (including up to 100,000 Hz) no matter its source. As mentioned in the introduction, there are many sources of ship noise below 1,000 Hz that should be expected to vary between individual ships in a particular class. Conversely, a wide range of vessels have been documented to radiate elevated high-frequency noise upon increased engine RPM or SOG—conditions reasonably associated with increased cavitation (Erbe & Farmer, 2000; Kipple, 2002; Hildebrand et al., 2006).

The literature offers a handful of spectra for particular classes that can be compared with the quantiles of Fig. 6. These spectra typically come from individual ships, though, so can only serve to verify the range of our quantiles, rather than assessing the accuracy of the quantiles themselves.

The spectrum levels provided by McKenna et al. (2012) for individual ships in comparable classes (a container ship, a vehicle carrier, two bulk carriers, and a few tankers) all fall within a few dB of our 95% quantile. Only their bulk carrier deviates from this pattern with levels near 100 Hz higher by about 10 dB. Overall, the broadband and spectrum levels of ships associated with the port of Los Angeles (McKenna et al., 2012) are most comparable to the noisiest 5% of ships transiting Haro Strait.

Similarly, the source spectrum levels for a single container ship measured in the middle of Haro Strait by Hildebrand et al. (2006) also fall within the 5% and 95% quantiles of our cargo class (from 90 Hz to 40,000 Hz). The alignment of such individual ship spectra within the quantiles of their associated class at all common frequencies—and most importantly at frequencies below that of our lowest transmission loss test tone—helps verify our extrapolation of the near-spherical spreading we observed from 630 to 20,100 Hz to all frequencies reported in our study.

We take this spectral consistency across multiple classes as evidence that the ship noise received at our nearshore hydrophone has not undergone shallow water attenuation. While normal mode theory (Urick, 1983) would predict a cutoff frequency of about 50 Hz if our hydrophone were in a shallow channel 8 m deep, that is not the bathymetric situation at our study site. Instead, Haro Strait is a 250–300 m deep channel with a steep western wall of sparsely sedimented solid rock (Jones & Wolfson, 2006) and our hydrophone is positioned near the top of the wall where the offshore bottom slope is 20–30°. In this situation, Jones & Wolfson (2006) expect not only destructive interference at ranges much greater than the source depth, but also upslope enhancement. In our transmission loss experiment, we did not observe any frequency dependent attenuation consistent with these phenomena. Furthermore, the theoretical cutoff frequency for a 250 m deep channel is 1.5 Hz (Urick, 1983), well below our lowest measured frequency band. We therefore argue that any effects of interference or backscatter are averaged out in our study, primarily because each isolated ship ensonifies the full width of this reverberating channel and moves 150–300 m during a 30-second recording (1–2 times the 130 m wavelength or our lowest measured frequency, 11.5 Hz).

Conclusions

Having ensured our samples were isolated (uncontaminated by noise from other ships or boats) and subtracted estimated background levels, we are confident that median received levels of ship noise in the core of SRKW critical habitat are elevated above median background levels not only at low frequencies (20–30 dB from 100 to 1,000 Hz), but also at high frequencies (5–13 dB from 10,000 to 40,000 Hz). Thus, underwater noise radiated by modern ships extends to high frequencies just as boat noise does (Erbe, 2002; Kipple & Gabriele, 2004; Hildebrand et al., 2006). Earlier studies have also observed this aspect of ship noise, but with smaller sample size, over different frequency ranges and less diverse ship classes (Kipple & Gabriele, 2004; Hildebrand et al., 2006; Bassett et al., 2012), and/or in received rather than source levels (Hermannsen et al., 2014).

Such ship noise has the potential to mask odontocete signals, especially in coastal environments where shipping lanes are close enough to the shoreline (<10 km) that high frequency sound is not fully absorbed. In the summertime habitat of the endangered SRKWs ship noise may interfere not only with SRKW communication (vocalizations) but also foraging and navigation (echolocation clicks).

Average broadband received levels (11.5–40,000 Hz) for the entire ship population are 110 ± 7 dB and ranged from 97 ± 10 dB for pleasure craft to 116 ± 4 dB for container ships. The range of RL for container ships (112–120 dB) show that levels received by SRKWs along the coastline at Lime Kiln from some container ships occasionally meet or exceed the 120 dB broadband threshold currently used by NOAA to define level B harassment from non-impulsive noise in the US

Ships northbound in Haro Strait exhibit moderate speeds with low variability (SOG of 7.3 ± 2.0 m/s or 14.1 ± 3.9 knots). Nevertheless, there is enough variation in speed across the whole population to reveal a linear relationship between received level and speed with a slope of +1.8 dB per m/s. This suggests a potential mitigation strategy for the average ship—slowing down—that has been recommended previously as an operational ship quieting option (Southall & Scholik-Schlomer, 2008). This strategy has other environmental benefits, like reducing collision risks, and is consistent with recent industry efforts to increase fuel efficiency (e.g., the “slow steaming” initiative of Maersk). For a passenger ship measured at speeds of 9–18 knots (4.6–9.3 m/s) during WWII Ross (1976) shows in Figure 8.19 that reducing speed lowers source spectrum levels by at about 1.5 dB/knot (2.9 dB per m/s) at all frequencies, but most notably lowers them by about 3.0 dB/knot (5.8 dB per m/s)—both at high frequencies (above 10,000 Hz) and at low frequencies (less than 100 Hz).

Average broadband source levels were 173 ± 7 dB for the population. Comparing broadband source levels between ship classes, container ships have the highest mean SL of 178 ± 4 dB. Therefore, assuming near-spherical transmission loss, marine life within a couple kilometers of shipping lanes will commonly receive noise levels above NOAA’s 120 dB threshold. At ranges less than about a kilometer, receive levels from many ships in Haro Strait will exceed the 130–150 dB modeled ship noise (10–50,000 Hz) dose associated with minor changes in northern resident killer whale behavior (Williams et al., 2014).

At distances of less than about a kilometer, it is likely that received 1/12- or 1/3-octave band levels at high frequencies are equal or greater than they are at low frequencies. Further research should measure ship spectrum levels at ranges of a few hundred meters in order to more fully quantify the high frequency (40,000–100,000 Hz) components of ship sound signatures.

Models of noise impacts in habitat containing shipping lanes will be more accurate if parameterized with spectral data, as opposed to broadband levels. Since we observe spectral variability between and within the 12 classes of vessels in this study, most prominently the bifurcation at low frequencies between classes, such models should use the class-specific spectrum level quantiles if possible, rather than the whole-population spectrum and band level medians we have presented.

Our broadband, spectrum, 1/12-octave, and 1/3-octave source levels for the whole population have median values that are comparable to the literature, with a few exceptions that we believe are due primarily to methodological differences. Some past analyses may not have made all recommended corrections (TC43 Acoustics, 2012); most commonly, methods sections are ambiguous about the definition and subtraction of background noise levels from total received levels prior to source level computations. It is also possible that these exceptions are due to sampling ship populations that are distinct (being composed of different individual ships/classes and/or operating differently). Even though our sample size is larger than most previous studies, we estimate that we sampled only about 1.6% of the 86,942 ships in the 2012/2013 global fleet (UNCTD, 2013). In any case, since our source level quantiles have slightly lower levels than some studies, particularly at low frequencies, they can be taken as a conservative characterization of the current fleet when developing ship noise models or policies.

One subtle pattern we note is that compared to some previous measurements and models, our median source spectrum levels are relatively low below 200 Hz and relatively high above 20,000 Hz. One implication of this is that noise models using previous measurements may overestimate the low-frequency noise levels of some ship types and underestimate high-frequency noise levels. Such flattening of the spectral slope in more modern ships is described in Figure 8.20 of Ross (1976) which shows source spectrum levels (below 100 Hz and from 1,000 to 20,000 Hz) elevated 1–3 dB in large populations of post-War versus WWII-era vessels. Some studies show a flattening of spectra above 100–1,000 Hz as ship and engine speed increases (Ross, 1976; Arveson & Vendittis, 2000; Kipple, 2002). We speculate that this historical trend may be continuing and recommend further investigation of the evolution of both ship speed (Leaper, Renilson & Ryan, 2014) and the mitigation of low-frequency internal noise on ships for human health reasons.

We recommend that future ship noise studies statistically characterize populations of ships—both their broadband and spectrum source levels. Having struggled to discern which studies in the literature are comparable to our results, we also suggest that future method sections be explicit about ship classification, calibration procedures, background subtraction and/or criteria for isolation from other sources, models and/or measurements of transmission loss, band width(s) and centers, absorption, and any other corrections. Metadata should include statistical representations of ship speeds and measurement ranges. Many studies are ambiguous about some of these details which complicates replication, comparison of results, and formation of hypotheses about observed differences.

Future work should also assess covariates other than speed, such as size, as well as azimuthal and temporal variability in source spectrum levels. We know from years of listening to live audio streams of Salish Sea ship noise (free via orcasound.net) that there is great temporal variability in the noise radiated by many ships. A small percentage of ships emit periodic strong mid-frequency tones that are likely caused by singing propellers (Ross, 1976). Our next step is to explore such temporal variations in amplitude and frequency, identify statistical outliers that may represent extreme masking cases, and further investigate possible governing variables, including speed, class, azimuth, and loading.

The variability we observe within ship classes indicates opportunities for reducing noise in ships, particularly those associated with the upper quantiles in each class. While the details of the spectral and temporal variability of noise from an individual ship may be important to a receiving species, metrics for measuring and regulating underwater noise will practically involve some temporal averaging, and possibly integration over bands wider than 1 Hz. We suggest a reasonable time scale for averaging ship noise is seconds or minutes, rather than a year as stipulated in the European Union’s Marine Strategy Framework Directive 2008/56/EC (Tasker et al., 2010). Additionally, based on the received signal above background noise that we observe at high frequencies, we recommend that future guidelines for monitoring ship noise raise the upper frequency limit of recording systems from 20,000 Hz (Dekeling et al., 2014) to at least 50,000 Hz. As Registered Ship Classification Societies continue to issue underwater radiated noise notations, we hope that these data can be used to assess their validity.

Supplemental Information

Supplemental Information 1 R files: processed data and plotting script

These related files should allow readers to re-create our plots and/or access the processed data underlying our plots.

Click here for additional data file.

We would like to thank all who helped deploy and maintain the calibrated hydrophone system. Logistical support was provided by The Whale Museum (Jenny Atkinson and Eric Eisenhardt), Beam Reach, intrepid divers (David Howitt), and SMRU Consulting. Analysis was accomplished through open-source software and data including: Generic Mapping Tools (GMT), NOAA bathymetry and shipping lanes, Qt, R, ggplot, Libreoffice, Overleaf, and Zotero. Chris Bassett and Marla Holt kindly provided helpful reviews of the pre-print; Michael Jasny and Hussein Alidina helped us understand the policy implications of our work; Leslie Veirs, Wendy Wood, and Annie Reese provided unflinching encouragement throughout. Finally, we thank the libraries of the University of Washington and Friday Harbor Labs for access to otherwise closed-access journals.

Additional Information and Declarations

Competing Interests

Author Contributions

Data Availability

1 All decibels here are referred to 1 µPa and source levels to a distance of 1 m. After their first usage, the units of broadband and spectrum level decibels are generally suppressed.

The authors declare there are no competing interests.

Scott Veirs and Val Veirs conceived and designed the experiments, performed the experiments, analyzed the data, contributed reagents/materials/analysis tools, wrote the paper, prepared figures and/or tables, and reviewed drafts of the paper.

Jason D. Wood conceived and designed the experiments, performed the experiments, analyzed the data, contributed reagents/materials/analysis tools, wrote the paper, and reviewed drafts of the paper.

The following information was supplied regarding data availability:

http://www.beamreach.org/data/staff-research/ship-noise/.

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
