# Peer review of "Ship noise extends to frequencies used for echolocation by endangered killer whales"

_PeerJ, doi:10.7717/peerj.1657_

## Round 0.1 · original submission · Minor Revisions

All three reviewers are very positive about this article but have suggested minor revisions, some of which are critical. I would ask you to consider whether it is appropriate to use means and standard deviations. Are your data normally distributed? If you are using medians and quartiles etc, it isn't necessary to present means as well. You mention a slope for the relationship between speed and noise - you should really give more details of how you arrived at that. If your data are not normally distributed you should use a non-parametric approach or just use non-parametric correlation tests what will tell you whether the relationship is positive/negative and whether it is significant. I hope I'm not teaching you to suck eggs!

Reviewer 1 ·

Basic reporting

No comments

Experimental design

No Comments

Validity of the findings

No comments

Additional comments

This article is extremely thorough in its reporting, and adheres to all PeerJ policies. Personally, I would have liked some more figures in order to illustrate the density of empirical data. I wonder if this is something the authors might consider, but it is not enough to stop me recommending this article for publication.

This article articulates its experimental design in an exemplary manner. The primary research is clear, and the methodology wholly appropriate. The investigation was conducted at a high technical standard, and could (should) become a model for further investigation across the globe.

The findings are rigorously challenged, and based on solid data sets. There are few areas of conflicting comparative data, which does not negate the findings.

·

Basic reporting

No comments

Experimental design

No comments

Validity of the findings

Propagation data taken above 500 Hz should not be used to explain propagation below 400 Hz. The broadband values would not be affected too much but the narrow band levels below 400 Hz are probably incorrect. However, relative differences (between ships) may be valid if the data are acquired at similar distances.

Source levels below 200 Hz are very dependent on source depth and bottom conditions. Source levels for shallower ships would be lower than those from deeper draft ships. this may explain some of the differences noted between ships.

The high frequency levels predicted after adjusting for absorption are not consistent with data taken close (within 300 m) to ships - just noting this in the text should suffice. Also, were absorption losses apparent in the propagation experiments?

Additional comments

Image (surface and bottom) contributions severely affect ship noise levels below 400 Hz. If the acquisition geometry is similar, relative comparisons may be valid. Source level estimations depend on a knowledge of source depth and are usually avoided.

·

Basic reporting

No comments

Experimental design

The authors could improve on "clearly defining the research question" by stating the main objectives in the introduction and then returning to these objectives in the Results/Discussion section. I make some specific recommendations in the "General Comments" section of the review.

Validity of the findings

No comments

Additional comments

1- The only major issue with the paper (which is actually pretty minor) is that when reporting a difference in dB, "re 1uPa/Hz" or "re 1uPa/Hz @1 m" should not be reported as these terms are dropped in a comparison between dB values. This error is found throughout the manuscript and needs to be corrected before publication- it will also shorten the paper quite a bit!
2- Not sure the official policy of the journal but a more standard unit for speed is m/s (not knots); however knots is more common in this field of research. Consider reporting the m/s with knots in parentheses and make this consistent throughout the manuscript
3- Consider making the title shorter "Ship noise extends to frequencies used for echolocation by endangered killer whales" ; the main reason for this suggestion is that high frequency ship noise is not unique to the location.
4- Consider adding a concluding sentence to the abstract about how these measurements of ship noise are unique- something like this is first study to characterize the noise from ships operating in coastal habitats and in frequencies relevant to killer whales. This study is a huge contribution- and some more concise writing will make that clear to any reader!
5- The introduction includes all the important background information to the study; however, some reorganization could improve the readability of the material and better highlight how this study is unique. Consider starting with a more concise paragraph about what we know about ship noise (low frequency component) and the implications for interference with long range communication of baleen whale communication. Then state how your study expands this understanding to include frequencies relevant to killer whale foraging and communication.
6- Consider including specific objective statements at the end of the intro- something like: First objective was to characterize noise from ships in higher frequencies and how this effects the frequencies important to killer whales; second, was to compare the measurements of this study with previous studies to understand consistences and possible biases with different field measurements. I think this will help readers connect the rest of the introduction (and the paper).
6- The first section of the methods might not be needed as the details are repeated in the sub-headings; possibly add some details to the introduction
7- line 254, how consistent are the oceanographic conditions in the region? Consider adding a statement about how representative the March measurement is across all months ships were measured
8- line 285, assume the SLa just indicates that SL was calculated with absorption? Might want to be explicit- it is on line 296, so maybe okay?
9- line 289, please provide a reference for ISO- what standard?
10- line 299, consider changing the title of the section to “Ship Traffic Patterns”
11- line 308, can you provide any information on the proportion of the global fleet that were measured in this study, particularly for the container and bulk carriers
12- This is sort of a general comment for the results and discussion section. When comparing with previous studies, important to summarize all the possible differences that could influence a RL and SL when collecting opportunistic measurements- most are mentioned but a bit scattered throughout the text. Perhaps a short section stating all the factors. I think this would be valuable so difference in measurements are clearly stated. If you do some reorganization based on the two main objectives, this could be the start of a section on the second objective suggested above.
13- line 376, The speed and distance ranges for McKenna 2013 are provided in the supplementary material: speed: mean = 10.5 m/s (SD 1.4) and ranges: mean = 3246 m (SD 291.2)

---

## Round 0.2 · accepted · Accept

Thank you for your very comprehensive response to the reviewers' comments. Having seen it in action through your submission, I will be investigating the use of "Overleaf" for my own work in future!